# The pronounced lung lesions developing in *LAT*^Y136F^ knock-in mice mimic human IgG4-related lung disease

Yuko Waseda[1]*, Kazunori Yamada[2], Keishi Mizuguchi[3], Kiyoaki Ito[2], Satoshi Watanabe[4], Masahiko Zuka[5], Tamotsu Ishizuka[1], Marie Malissen[6], Bernard Malissen[6], Mitsuhiro Kawano[2], Shoko Matsui[7]

1 Third Department of Internal Medicine, Faculty of Medical Sciences, University of Fukui, Eiheiji, Fukui, Japan, 2 Department of Rheumatology, Kanazawa University Hospital, Kanazawa, Japan, 3 Department of Diagnostic Pathology, Kanazawa University Hospital, Kanazawa, Japan, 4 Department oh Respiratory Medicine, Kanazawa University Hospital, Kanazawa, Japan, 5 Department of Forensic Medicine and Pathology, Kanazawa University Graduate School of Medical Sciences, Kanazawa, Japan, 6 Centre d'Immunologie de Marseille-Luminy, Aix Marseille Universite´, INSERM, CNRS, Marseille, France, 7 Health Administration Center, University of Toyama, Toyama, Japan

* yuwaseda@gmail.com

**Data Availability Statement:** All relevant data are within the manuscript and its Supporting information files.

## Abstract

### Rationale

Immunoglobulin (Ig) G4-related disease (IgG4-RD) is a novel clinical disease entity characterized by an elevated serum IgG4 concentration and tumefaction or tissue infiltration by IgG4-positive plasma cells. Pathological changes are most frequently seen in the pancreas, lacrimal glands, and salivary glands, but pathological changes in the lung also exist. Linker for activation of T cell (*LAT*)^Y136F^ knock-in mice show Th2-dominant immunoreactions with elevated serum IgG1 levels, corresponding to human IgG4. We have reported that *LAT*^Y136F^ knock-in mice display several characteristic features of IgG4-RD and concluded that they constitute an appropriate model of human IgG4-RD in salivary glands, pancreas, and kidney lesions.

### Objectives

The aim of this study is to evaluate whether lung lesions in *LAT*^Y136F^ knock-in mice can be a model of IgG4-related lung disease.

### Methods

Lung tissue samples from *LAT*^Y136F^ knock-in mice (LAT) and wild-type mice (WT) were immunostained for IgG1 and obtained for pathological evaluation, and cell fractions and cytokine levels in broncho-alveolar lavage fluid (BALF) were analyzed.

### Results

In the LAT group, IgG1-positive inflammatory cells increased starting at 4 weeks of age and peaked at 10 weeks of age. The total cell count and percentage of lymphocytes increased

**Funding:** Japan society for the promotion of science Grants-in-Aid for Scientific Research Yuko Waseda: JSPS KAKENHI (19K08646, 16K09531) https://nrid.nii.ac.jp/ja/nrid/1000080536037/ Kazunori Yamada: JSPS KAKENHI (17K09999) https://nrid.nii.ac.jp/ja/nrid/1000090397224/ Mitsuhiro Kawano: JSPS KAKENHI (26461487) https://nrid.nii.ac.jp/ja/nrid/1000020361983/ Shoko Matsui: JSPS KAKENHI (19K08645, 15K09212, 24591158) https://nrid.nii.ac.jp/ja/nrid/1000040334726/.

**Competing interests:** The authors have declared that no competing interests exist.

significantly in BALF in the LAT group compared to the WT group. In BALF, Th2-dominant cytokines and transforming growth factor-β were also increased. In the LAT group, marked inflammation around broncho-vascular bundles peaked at 10 weeks of age. After 10 weeks, fibrosis around broncho-vascular bundles and bronchiectasis were observed in $LAT^{Y136F}$ knock-in mice but not WT mice.

## Conclusions

$LAT^{Y136F}$ knock-in mice constitute an appropriate model of lung lesions in IgG4-RD.

## Introduction

In recent years, immunoglobulin (Ig) G4-related disease (IgG4-RD) has been established as a new disease entity [1]. IgG4-RD is characterized by high levels of IgG and IgG4 in serum, infiltration of IgG4-positive plasma cells, obliterative phlebitis or obliterative arteritis and lymphocytes, as well as a form of fibrosis called storiform fibrosis in some organs, such as the pancreas, salivary glands, kidney, retroperitoneum, and periaorta [2–4].

IgG4 is a subclass of IgG without complement binding ability and is associated with a decrease in symptoms in the context of IgE-mediated allergy due to an allergen-blocking effect at the mast cell level and/or at the level of the antigen-presenting cell [5]. This means that IgG4-RD has IgE-mediated type I allergy as a characteristic of the disease, and IgG4 as a blocking antibody may accumulate in each organ.

IgG4-RD is said to coexist with many allergic diseases, such as rhinitis, conjunctivitis, bronchial asthma, and urticaria [6]. Interleukins (IL) 4, 5, 10, and 13 and transforming factor β (TGF-β) are overexpressed through an immune reaction in which type 2 helper T cells (Th2) predominate, followed by activation of regulatory T cells [7]. These cytokines contribute to eosinophilia, elevated serum IgG4 and IgE concentrations, and progression of fibrosis that are characteristic of IgG4-RD [8].

In humans with IgG4-RD, some clinicoradiological and pathological studies of IgG4-related lung disease (IgG4-RLD) demonstrated that intrathoracic lesions include lymphatic routes with vascular involvement [9–11].

Regarding the lungs, IgG4-positive plasma cells may infiltrate into the lung even in connective tissue diseases [12], and patients with diseases other than IgG4-RD will be included based on comprehensive diagnostic criteria alone. Therefore, we established diagnostic criteria unique to the lungs so that lung lesions can be accurately diagnosed based on clinical, radiological, and pathological findings [13].

IgG4-RLD is also likely to be associated with an allergic disease. Recently, several studies have been conducted on the cause of IgG4-RD [14, 15]. However, the relationship between these diseases and Th2-dominated allergic reactions, including reactions in the lung, is not clear.

Linker for activation of T cell (LAT) is an adaptor protein that is a major substrate for the Zap70 protein tyrosine kinase in T cells and initiates most intracellular events that characterize the T cell receptor signaling pathway [16]. In $LAT^{Y136F}$ mice, in which the tyrosine residue at position 136 is replaced with phenylalanine, binding of LAT to phospholipase C-γ1 is lost. $LAT^{Y136F}$ knock-in mice develop lymphoproliferative disorder and show expansion of polyclonal CD4+ T cells along with high Th2 cytokine production. In parallel, they show massive activation of B cells as well as elevated serum levels of IgG1 and IgE. Mouse IgG1 is induced by

Th2 cytokines and does not bind to C1q. Thus, mouse IgG1 is considered to be a homologue of human IgG4 [17, 18].

Several animal models of IgG4-RD have been established [19–22]. We first reported that *LAT*$^{Y136F}$ knock-in mice display several immunopathological features characteristic of IgG4-RD in organs such as salivary glands, pancreas, and kidney, as well as dural lesions in which many IgG1-positive cells and fibrosis are present [23, 24].

Because IgG4-RD shows many intrathoracic lesions on chest computed tomography, and allergic symptoms are also observed, we examined whether *LAT*$^{Y136F}$ knock-in mice constitute a model of the lung pathology associated with IgG4-RLD. Thus, the aim of this study was to estimate lung lesions in *LAT*$^{Y136F}$ knock-in mice and evaluate whether *LAT*$^{Y136F}$ knock-in mice constitute an appropriate model of human IgG4-RLD.

## Methods

### Animals

All experimental procedures were reviewed and approved by the Animal Experimentation Committee and Gene Recombination Experiment Safety Committee of Kanazawa University (AP-10174, Kindai6-1013) from 2013 to 2019. We used a combination of midazolam, medetomidine, and butorphanol tartrate for anesthesia to minimize discomfort to the mice. For euthanasia, we placed the mice in a closed container filled with carbon dioxide gas, exposed the mice for a sufficient period of time, confirmed complete respiratory arrest, and then performed cervical dislocation. *LAT*$^{Y136F}$ knock-in mice [17] and WT C57BL/6 mice were housed in a controlled temperature (23 ± 2˚C) and humidity (30–70%), and circadian light-dark cycle of 12 h, with ad libitum access to food and water and underwent microbiological monitoring tests four times a year.

All animal experiments were performed in accordance with the Guidelines for the Care and Use of Laboratory Animals of the Takara-machi Campus of Kanazawa University.

### Study population

*LAT*$^{Y136F}$ knock-in mice (n = 66; males, 36; females, 30) and wild-type (WT) mice (n = 70; males, 38; females, 32) 4–20 weeks old were included in this study.

### Histopathology and immunohistochemistry

Because we observed that the development of different tissue lesions was comparable in male and female mice, mice of both sexes were used for the present study. Tissue samples were fixed in 20% formalin and embedded in paraffin. To score the largest cut surface over the entire field of view, all organs were sectioned and stained with hematoxylin-eosin (H-E) for basic observations, with Elastica van Gieson (EVG) stain for elastic fibers, and with Azan for estimation of collagen in fibrosis. The specific primary antibodies used were as follows: goat anti-mouse IgG1 (1:2000, SouthernBiotech, Birmingham, AL, USA), goat anti-mouse IgG (1:4000, SouthernBiotech), and rat purified anti-mouse CD138 (syndecan-1; 1:1000, BioLegend, San Diego, CA, USA). Heat-induced epitope retrieval was performed at 120˚C for 10 min using 10 mM citrate buffer (pH 6) without Tween 20. Ethylenediamine-tetraacetic acid buffer (pH 9) was also used for CD138. Thereafter, the tissues were incubated with the primary antibodies at 4˚C overnight. After washing with phosphate-buffered saline, the sections were incubated with an amino acid polymer (Hystofine simple stain mouse MAX-PO (R), Nichirei, Tokyo, Japan) at room temperature for 30 min according to the manufacturer's instructions. After washing

with phosphate-buffered saline again, the sections were reacted with diaminobenzidine-tetra-hydrochloride solution (DAKO, Glostrup, Denmark).

The severity of fibrosis was semiquantitatively assessed according to the method of Ashcroft et al. [25] The entire lung section was viewed at a magnification of 100×. Thirty to thirty-five fields in each section were analyzed, and a score ranging from 0 (normal lung) to 8 (total fibrosis) was assigned. The mean score of all fields was taken as the fibrosis score of that lung section. Criteria for grading pulmonary fibrosis were as follows. Grade 0 = normal lung; Grade 1 = minimal fibrous thickening of alveolar or bronchial walls; Grade 2–3 = moderate thickening of walls without obvious damage to lung architecture; Grade 4–5 = increased fibrous mass; Grade 6–7 = severe distortion of structure and large fibrous areas; Grade 8 = total fibrotic obliteration of the field. CD138+ and IgG1+ cells were counted under a 200× objective. Evaluation of tissue specimens was performed by one cytotechnologist and one pulmonologist who were blinded to the genotype of the mice.

## Broncho-alveolar lavage (BAL)

*LAT*<sup>*Y136F*</sup> knock-in mice and WT mice were killed at 8–9 weeks of age, and BAL was obtained. After excision of the trachea, a plastic cannula was inserted into the trachea, and 2 ml saline solution was injected gently with a syringe and then withdrawn. This procedure was repeated three times. A 100-μl aliquot of BAL fluid (BALF) was reserved for total cell counts and evaluation of cell differentiation. The remaining BALF was centrifuged immediately at 1,100 rpm for 10 min. The total cell number was determined with a standard hemocytometer. Cell differentiation was examined by counting at least 200 cells on smears prepared with cytospin and Wright-Giemsa staining. Supernatants were stored at −80°C until use for measurement of cytokines.

## Measurement of cytokines/chemokines in BALF

The remaining BALF was used to measure levels of the following cytokines: IL-4, IL-5, IL-6, IL-10, IL-13, interferon (IFN)-γ, tumor necrosis factor (TNF)-α, eotaxin, and RANTES (Luminex® Multiplex Assays; ThermoFisher Scientific/Invitrogen, Tokyo, Japan). TGF-β1 was measured with an enzyme-linked immunosorbent assay (Quantikine; R&D Systems, Minneapolis, MN, USA).

## Statistical analysis

Mann-Whitney U tests were performed for comparison of the Ashcroft score and cytokines and chemokines in BALF. Statistical analysis was performed using statistical software (SPSS 12.0J, 2003, SPSS Inc., Chicago, IL, USA).

## Results

### Development of tissue inflammatory lesions in the lung of *LAT*<sup>*Y136F*</sup> knock-in mice starts with infiltration of IgG1-positive lymphoplasmacytic cells and is followed by fibrosis

H-E staining showed that the lungs were almost normal at 4 weeks of age, and inflammation around the bronchi, bronchioles, and vessels began after 4 weeks of age. Inflammation peaked at 6–10 weeks and decreased at 20 weeks. Fibrosis was worse and resembled traction bronchiectasis at 20 weeks. Fig 1, lower left, shows a 20-week lung that was also bleeding.

# Pathological findings

**Fig 1. Histopathological appearance of the lung of *LAT*<sup>Y136F</sup> knock-in mice.** Hematoxylin-eosin (H-E) staining showed that inflammation started after 4 weeks and peaked at 6–10 weeks. Inflammation decreased at 20 weeks, but the fibrosis was getting worse.

Six-week lung tissue showed marked lymphoplasmacytic cell infiltration into the interstitium of the peribronchovascular wall with H-E staining, obliterative phlebitis with EVG staining, and fibrosis with Azan staining (Fig 2).

In the LAT group, the numbers of CD138+ and IgG1+ cells were 300 and 240 cells/high-power field at 10 weeks, respectively. The cell numbers peaked at 10 weeks and then decreased. In contrast, in the WT group, neither CD138+ nor IgG1+ cells changed throughout this time course (Fig 3A and 3B).

When fibrosis was evaluated using the Ashcroft score, we found no difference between *LAT*<sup>Y136F</sup> knock-in mice and WT mice at 4 weeks, but *LAT*<sup>Y136F</sup> knock-in mice had a significantly higher score than WT mice after 6 weeks (LAT vs. WT at 6 w: 4.5 vs. 0.6, $p < 0.05$, at 10

## Pathological findings

**Fig 2. Inflammatory cells, obliterative phlebitis, and storiform fibrosis existed in *LAT^{Y136F}* knock-in mice.** H-E staining in lung tissues at 6 weeks of age. a, b; inflammatory cells around the bronchi (a: H-E ×40, b: H-E ×200, enlargement of the boxed area in a), c; obliterative phlebitis (EVG ×400), d; fibrosis (Azan ×200).

w: 4.9 vs. 0.4, $p < 0.05$, and at 20 w: 5.8 vs. 0.6, $p < 0.05$). In addition, in *LAT^{Y136F}* knock-in mice, the score significantly increased over time (LAT 6 w vs. LAT 10 w: 4.5 vs. 4.9, $p < 0.05$, LAT 10 w vs. LAT 20 w: 4.9 vs. 5.8, $p < 0.05$). Therefore, lung fibrosis began at 4 weeks and worsened at 20 weeks in the LAT group (Fig 4).

## BALF findings in *LAT^{Y136F}* knock-in mice with increased lymphocytes and Th2-dominant cytokines

To evaluate the cellular and humoral components of BALF at the time when a large amount of cell infiltration was present in the tissue, BALF of 8- to 9-week-old mice was evaluated after the tissue evaluation.

Regarding BALF cell composition, the lymphocyte percentage found in the LAT group was higher than in the WT group ($74.5 \pm 11.0$ vs. $15.5 \pm 17.2$, respectively, $p < 0.01$). Moreover, the total cell count and numbers of lymphocytes were significantly higher in the LAT group than

(A) **Pathological findings**

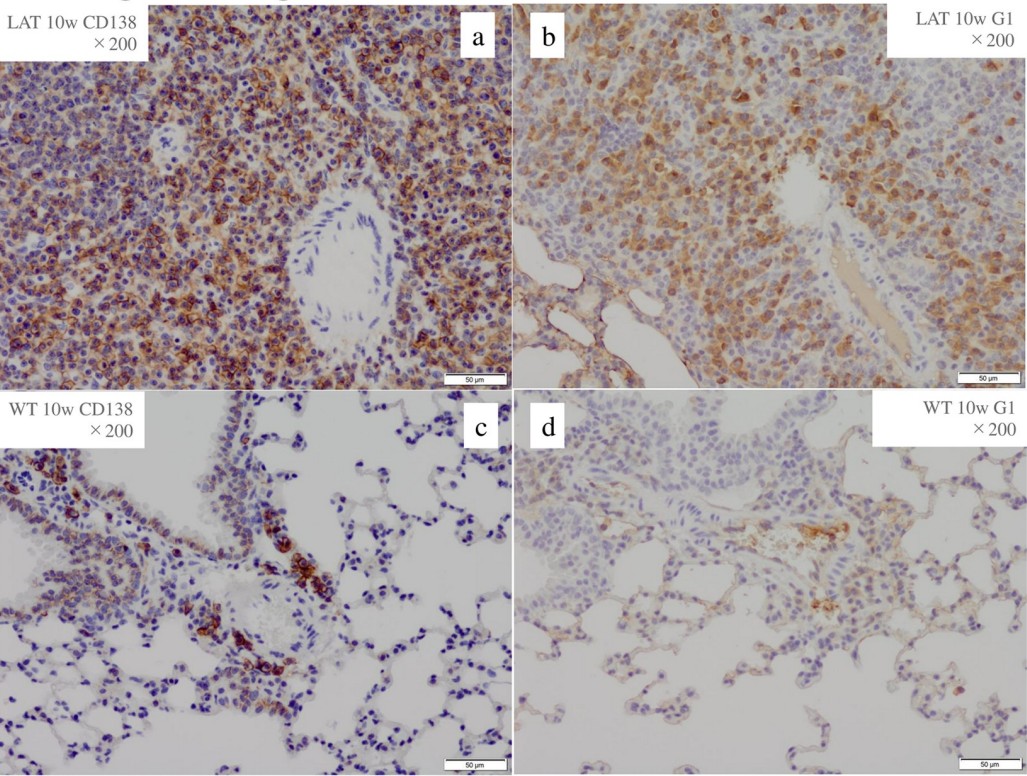

(B) **Pathological findings**

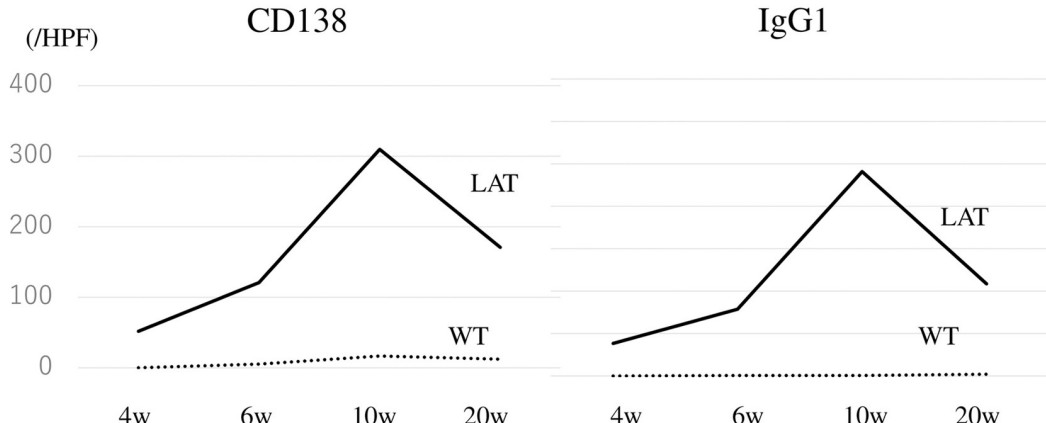

**Fig 3. Immunostaining for plasma cells in tissue lesions of 10-week-old *LAT*<sup>Y136F</sup> knock-in mice.** In the LAT group, the numbers of CD138+ and IgG1+ cells peaked at 10 weeks and then decreased (3A a, b, 3B). In the WT group, neither CD138 + cells nor IgG1+ cells changed (3A c, d, 3B).

## Achcroft score

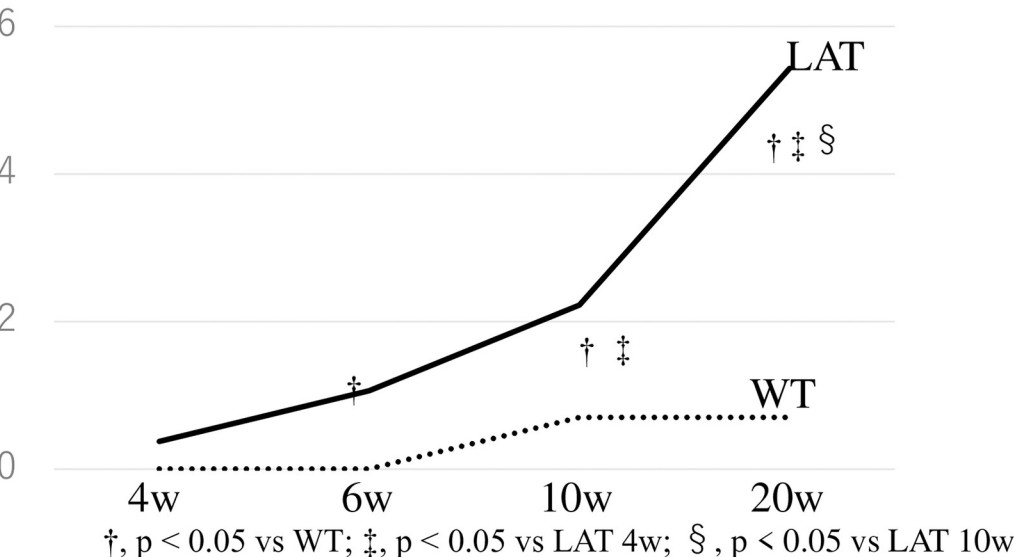

**Fig 4. Evaluation of fibrosis according to the Ashcroft score in *LAT^{Y136F}* knock-in mice.** When fibrosis was evaluated using the Ashcroft score, lung fibrosis began at the 4th week and was significantly getting worse at 20 weeks in the LAT group. On the other hand, no significant change was seen in the WT group.

the WT group (11.9 ± 8.1 vs. 2.0 ± 0.4, p < 0.01; 10.4 ± 7.3 vs. 0.42 ± 0.37, p < 0.01, respectively) (Fig 5).

In the LAT group, the BALF levels of IL-13 and eotaxin were significantly higher (IL-13: 0.80 ± 0.8 vs. 0.47 ± 0.2, p < 0.01, eotaxin: 48.9 ± 22.5 vs. 3.8 ± 1.5, p < 0.01), and the IFN-γ levels were significantly lower than in the WT group (0.06 ± 0.02 vs. 0.4 ± 2.6, p < 0.005). IL-4 also tended to be higher in the LAT group than the WT group (2.9 ± 2.6 vs. 1.4 ± 0.7, p = 0.081).

IL-10 was not significantly different between the two groups (LAT group vs. WT group; 1.18 ± 6.1 vs. 2.3 ± 1.5, p = 0.271), and IL-5 and RANTES were below the level of sensitivity in both groups. TGF-β1, which may play a central role in the progression of fibrosis, was overexpressed in the LAT group compared to the WT group (207.7 ± 238.6 vs. 64.9 ± 32.5, p < 0.01). Likewise, TNF-α was significantly higher in the LAT group than in the WT group (3.2 ± 3.9 vs. 1.0 ± 1.2, respectively, p < 0.05) (Table 1).

## Discussion

IgG4-RD includes lesions with increased IgG4+ plasma cells, Th2 cytokine-producing CD4 + T cells, and regulatory T cells in multiple organs [5, 8]. The lesions are more common in the pancreas, lacrimal glands, and salivary glands, but lesions are also seen in the kidneys, bile ducts, and lungs [4].

## BALF findings

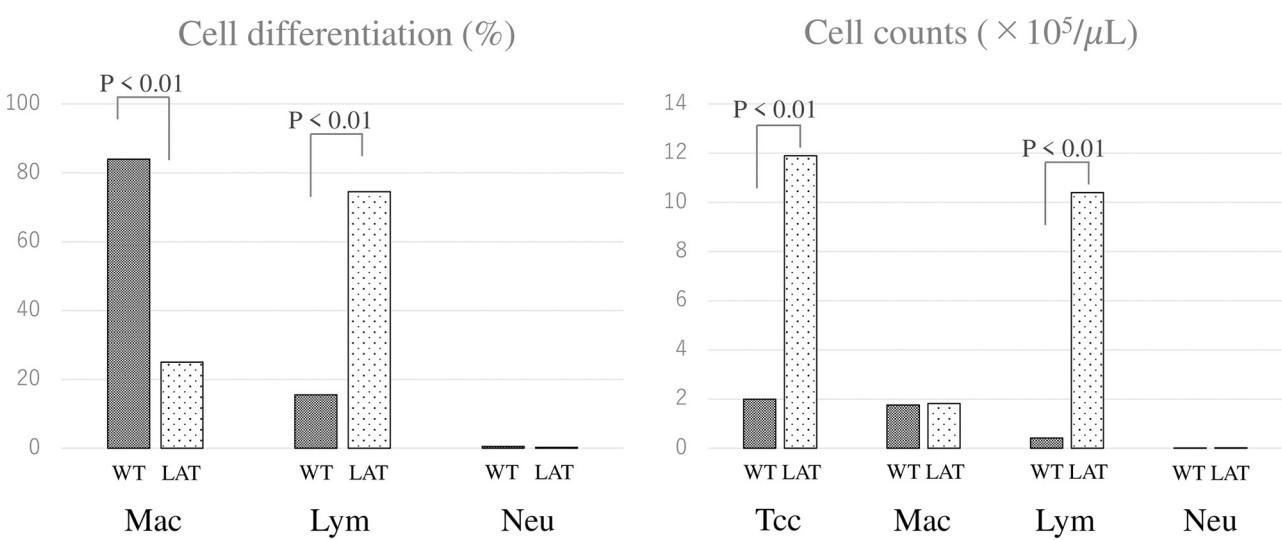

**Fig 5. Cell differentiation and cell counts of BALF in *LAT*<sup>Y136F</sup> knock-in mice.** Regarding BALF cell differentiation in the LAT group at 8–9 weeks of age, the lymphocyte fraction was predominantly higher than in the WT group. The total cell count and numbers of lymphocytes were significantly higher in the LAT group.

The histological findings of IgG4-RLD are marked lymphoplasmacytic cell infiltration into the interstitium of the peribronchovascular sheath, interlobular septal wall and/or pleura, an IgG4/IgG-positive cell ratio >40% and/or >10 IgG4-positive cells/high-power field, oblitera-tive phlebitis or obliterative arteritis, and storiform fibrosis or fibrosis consisting of proliferat-ing spindle-shaped cells around infiltrating lymphocytes [13].

**Table 1. Cytokine/chemokine findings in BALF.**

| | | LAT (n = 8) | WT (n = 7) | |
|---|---|---|---|---|
| IL-4 | (pg/mL) | 2.87 ± 2.55 | 1.35 ± 0.67 | p = 0.081 |
| IL-5 | (pg/mL) | 0.86 ± 0.76 | 0.65 ± 0.64 | p = 0.601 |
| IL-6 | (pg/mL) | 1.38 ± 1.41 | 1.01 ± 0.92 | p = 0.352 |
| IL-10 | (pg/mL) | 1.18 ± 6.05 | 2.27 ± 1.45 | p = 0.271 |
| IL-12p70 | (pg/mL) | 0.16 ± 0.03 | 0.16 ± 0.09 | p = 0.719 |
| IL-13 | (pg/mL) | 0.80 ± 0.82 | 0.47 ± 0.15 | p = 0.005 |
| IFN-γ | (pg/mL) | 0.06 ± 0.02 | 0.37 ± 2.64 | p = 0.001 |
| TNF-α | (pg/mL) | 3.23 ± 3.92 | 0.97 ± 1.15 | p = 0.028 |
| Eotaxin | (pg/mL) | 48.9 ± 22.5 | 3.77 ± 1.54 | p = 0.001 |
| Rantes | (pg/mL) | 0.82 ± 0.31 | 0.82 ± 0.31 | p = 0.896 |
| TGF-β1 | (pg/mL) | 207.7 ± 238.6 | 64.9 ± 32.5 | p = 0.003 |

In the LAT group, IL-13, TNF-α, eotaxin, and TGF-β1 levels in BALF were significantly higher, and the IFN-γ levels were significantly lower than in the WT group. TGF-β1, which may play a central role in promotion of fibrosis, was overexpressed in the LAT group. IL-10 was not significantly different between the two groups, and IL-5 and RANTES were below the level of sensitivity in both groups.

Several animal models of IgG4-RD have been reported [19–22]. Our previous study using *LAT*^Y136F knock-in mice examined lesions in the pancreas, salivary glands, and kidneys, as well as dural lesions [23]. Here, we further investigated whether these mice could also constitute appropriate models of lung lesions in IgG4-RD.

Because human IgG4 is considered to be equivalent to mouse IgG1, immunostaining for IgG and IgG1 in pathological tissues was performed as an indicator of inflammation [26]. IgG could not be stained well, and thus CD138, a marker of plasma cells, was used instead. Results showed that a large number of CD138+ and IgG1+ plasma cells were present in lung lesions, and the number increased significantly compared to WT mice and peaked at 10 weeks of age. Furthermore, the Th2 cytokines, IL-13 and IL-4, tended to be elevated in the BALF of *LAT*^Y136F knock-in mice. IL-5 was below the level of detection in the multiplex cytokine assay and could not be evaluated. In addition, the Th1 cytokine, IFN-γ, was low, which was consistent with the fact that these mice tended to express Th2 cytokines.

Regarding fibrosis, the value of TGF-β1 measured in BALF of *LAT*^Y136F knock-in mice was significantly higher than that in WT mice. In addition, in lung tissue, *LAT*^Y136F knock-in mice showed a significantly higher Ashcroft score, which is an indicator of fibrosis, than WT mice, indicating that fibrosis increased with time. Lung lesions of *LAT*^Y136F knock-in mice also had characteristic pathological findings of IgG4-RLD. This was similar to the findings previously described in other organs such as salivary glands, pancreas, and kidney in *LAT*^Y136F knock-in mice [23], indicating that lung lesions in *LAT*^Y136F knock-in mice are also very similar to IgG4-RD.

Interestingly, the inflammatory cytokine, TNF-α, was elevated in *LAT*^Y136F knock-in mice. Mast cells activated by IgE release TNF-α and cause systemic allergic inflammation [27], and the same may occur in *LAT*^Y136F knock-in mice. The role of mast cells in *LAT*^Y136F knock-in mice and their association with IgG4-RD should be explored in future studies.

Several cytokines and chemokines such as IL-5 and RANTES could not be detected in BALF using the multiplex cytokine assay. These are major Th2 cytokines, and more sensitive measurement methods are needed. Because BALF was diluted with physiological saline, the concentrations of certain cytokines may be low to undetectable. This result is considered to be a limitation of the present study, and measuring cytokines and chemokines in lung tissue remains to be performed in the future.

In conclusion, lung lesions in *LAT*^Y136F knock-in mice constitute an appropriate model of lung lesions in human IgG4-RD. This new and clinically relevant mouse model of IgG4-RD allows more detailed understanding of the pathogenesis and relevant therapeutic strategies. Further studies on the detailed cellular and molecular profiles that may accelerate inflammation and fibrosis of the lung are warranted.

## Supporting information

**S1 File.**
(DOCX)

## Author Contributions

**Conceptualization:** Yuko Waseda, Kazunori Yamada, Mitsuhiro Kawano, Shoko Matsui.

**Data curation:** Yuko Waseda, Keishi Mizuguchi, Kiyoaki Ito, Satoshi Watanabe, Masahiko Zuka.

**Formal analysis:** Yuko Waseda.

**Funding acquisition:** Yuko Waseda, Kazunori Yamada, Mitsuhiro Kawano, Shoko Matsui.

**Investigation:** Yuko Waseda, Kazunori Yamada, Keishi Mizuguchi, Kiyoaki Ito, Satoshi Watanabe, Mitsuhiro Kawano.

**Methodology:** Marie Malissen, Bernard Malissen.

**Project administration:** Yuko Waseda, Mitsuhiro Kawano, Shoko Matsui.

**Resources:** Yuko Waseda, Kazunori Yamada, Mitsuhiro Kawano, Shoko Matsui.

**Validation:** Yuko Waseda, Kazunori Yamada, Shoko Matsui.

**Writing – original draft:** Yuko Waseda.

**Writing – review & editing:** Yuko Waseda, Kazunori Yamada, Keishi Mizuguchi, Kiyoaki Ito, Satoshi Watanabe, Masahiko Zuka, Tamotsu Ishizuka, Mitsuhiro Kawano, Shoko Matsui.

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
