## [Decision Letter · Decision Letter 0]

16 Dec 2020

PONE-D-20-29792

The pronounced lung lesions developing in LATY136F knock-in mice mimic human IgG4-related lung disease

PLOS ONE

Dear Dr. Waseda,

Thank you for submitting your manuscript to PLOS ONE. After careful consideration, we feel that it has merit but does not fully meet PLOS ONE’s publication criteria as it currently stands. Therefore, we invite you to submit a revised version of the manuscript that addresses the points raised during the review process.

We greatly appreciate your patience during the very long time it took us to review your work. Both reviewers recommended minor revisions and I concur entirely with their decision.

We look forward to receiving your revised manuscript.

Kind regards,

Michal A Olszewski, DVM, PhD

Academic Editor

PLOS ONE

Journal Requirements:

2. At this time, we request that you  please report additional details in your Methods section regarding animal care, as per our editorial guidelines:

(i) Please state the source of mice used in the study (i.e. where the mice were originally purchased from)

(ii) Please describe the care received by the animals, including the frequency of monitoring and the criteria used to assess animal health and well-being.

(iii) Please confirm at what age the mice were euthanised.

Thank you for your attention to these requests.

3. Please provide the product number and any lot numbers of the antibodies, luminex multiplex assays and ELISA assays for your study.

4. To comply with PLOS ONE submission guidelines, in your Methods section, please provide additional information regarding your statistical analyses. For more information on PLOS ONE's expectations for statistical reporting, please see https://journals.plos.org/plosone/s/submission-guidelines.#loc-statistical-reporting

Editor's Comment

Apart from the clarifications requested by the reviewers, please make sure that the paper is edited by a native English speaker or a English editing company to help with fixing all grammar errors that make some areas of this paper unclear.

Reviewers' comments:

Reviewer's Responses to Questions

**Comments to the Author**

1. Is the manuscript technically sound, and do the data support the conclusions?

Reviewer #1: Yes

Reviewer #2: Partly

2. Has the statistical analysis been performed appropriately and rigorously? 

Reviewer #1: Yes

Reviewer #2: Yes

3. Have the authors made all data underlying the findings in their manuscript fully available?

Reviewer #1: Yes

Reviewer #2: Yes

4. Is the manuscript presented in an intelligible fashion and written in standard English?

Reviewer #1: Yes

Reviewer #2: No

5. Review Comments to the Author

Reviewer #1: The manuscript “The pronounced lung lesions developing in LATY136F knock-in mice mimic human IgG4-related lung disease” by Waseda et al is a concise report that correlated mice that express an alternate form of LAT and the spontaneous lymphoproliferative disorder they develop to human IgG4-related disease. The authors find that these mice exhibit strong Th2 inflammation early in life, followed by fibrosis. This study is quite straight-forward, but would benefit from more detail in many places.

Concerns:

1. This study uses knock-in of LAT in which tyrosine 136 is substituted with a phenylalanine, thereby preventing downstream signaling. This results in drastic lymphoproliferation. Is there any evidence that LAT is involved in human disease? The authors also discuss lung involvement in the introduction and mention clinical characteristics, but these characteristics are not defined. More detail about the clinical observations in this disease and how they correlate to what is seen in the animal model would be helpful.

2. The authors briefly mention the Th2 nature of their observations and the potential link with allergic disease, but there is no follow-up on this in either the experiments or in the discussion. Are there differences in IgE in this model?

3. Figure 1 needs more detail in the results and figure legend, especially regarding the 20 week sections. It seems that the image on the left is H&E staining and the image on the right is not, but this is unclear and the stains for each section needs to be defined.

4. The Ashcroft score needs to be defined.

5. In general, Figures 1-3 would benefit from greater details in the results section.

Reviewer #2: Dr. Wasada et al present a well argued manuscript describing the use of a LAT knock-in mouse as a model of IgG4-RD of the lung. This is a well executed follow-up study of their prior work, published in this journal, establishing this same mouse model in other organs commonly affected by human IgG4 disease. While the authors provide a compelling argument, there are a few minor issues that I suspect, if addressed, would improve the author's argument.

1) Figure 1: you present two LAT 20w x20 figures at the bottom with largely different appearances. Does this speak to the heterogeneity of the results at 20w? If so, would it help to show cumulative data, compiling the fibrosis scores. If this is true heterogeneity, it may be beneficial to add actual datapoints to Figure 3B, 4, and 5 instead of just showing the median.

2) Redundancy: the statement, "In addition, in Lat knock in mice, the score significantly increased over time." This is already shown in the sentence above, just with data re-arranged. I would re-word or eliminate this.

3) I'm assuming you chose to study BALF at 8,9 weeks due to your prior results suggesting peak inflammation occurred between 6-10 weeks. I wonder if you would have found greater cytokine production at an earlier time-point. It would have been interesting to see the composition of the lymphocytes found in the BALF. This could be done fairly easily using flow cytometry and could provide some insight into the unique cytokine pattern you found with bead array.

4) Next steps. Now that you have established this model, how do you intend to study it further?

5) Wording. There are some phrases throughout the manuscript that are hard to follow, or do not seem to add anything to your argument. For example, on page two, "Regarding the lungs, in addition to comprehensive diagnostic criteria, diagnostic criteria unique to the lungs were established, and diagnosis of lung lesions is performed based on the clinical characteristics." The clarity of your message could be improved greatly by a grammatical review focused on reducing redundancy.

6. PLOS authors have the option to publish the peer review history of their article (what does this mean?). If published, this will include your full peer review and any attached files.

Reviewer #1: No

Reviewer #2: No

---

## [Author Response · Author response to Decision Letter 0]

19 Jan 2021

January 12, 2021

Joerg Heber, Editor-in-Chief

PLOS ONE

Dear Dr. Heber:

We wish to re-submit our manuscript titled “The pronounced lung lesions developing in LATY136F knock-in mice mimic human IgG4-related lung disease”.

We thank you and the reviewers for your thoughtful suggestions and insights. The manuscript has benefited from these insightful suggestions. I look forward to working with you and the reviewers to move this manuscript closer to publication in PLOS ONE.

The manuscript has been rechecked, and the necessary changes have been made in accordance with the reviewers’ suggestions. The responses to all comments are given below. 

Thank you for your consideration. I look forward to hearing from you.

Sincerely,

Yuko Waseda

Third Department of Internal Medicine, Faculty of Medical Sciences, University of Fukui, 23-3 Matsuoka Shimoaizuki, Eiheiji, Fukui, Japan, 910-1193

TEL: +81-776-61-8355

FAX: +81-776-61-8111

E-mail: yuwaseda@gmail.com

 

RESPONSE TO REVIEWER’S COMMENTS

Reviewer #1: The manuscript “The pronounced lung lesions developing in LATY136F knock-in mice mimic human IgG4-related lung disease” by Waseda et al is a concise report that correlated mice that express an alternate form of LAT and the spontaneous lymphoproliferative disorder they develop to human IgG4-related disease. The authors find that these mice exhibit strong Th2 inflammation early in life, followed by fibrosis. This study is quite straight-forward, but would benefit from more detail in many places.

Concerns:

1. This study uses knock-in of LAT in which tyrosine 136 is substituted with a phenylalanine, thereby preventing downstream signaling. This results in drastic lymphoproliferation. Is there any evidence that LAT is involved in human disease? The authors also discuss lung involvement in the introduction and mention clinical characteristics, but these characteristics are not defined. More detail about the clinical observations in this disease and how they correlate to what is seen in the animal model would be helpful.

COMMENT: 1

Thank you for your comment. First, whether LAT is related to human disease is unclear. The characteristics of lung lesions in human IgG4-RD were described in the Introduction. The pathological findings include high levels of IgG and IgG4 in serum, infiltration of IgG4-positive plasma cells, obliterative phlebitis or obliterative arteritis and lymphocytes, as well as a form of fibrosis called storiform fibrosis in some organs, such as the pancreas, salivary glands, kidney, retroperitoneum, and periaorta. We found that the lungs of LAT mice also had high serum IgG and IgG1 levels and had histopathology similar to that of humans. Clinical findings include many allergic diseases, such as rhinitis, conjunctivitis, bronchial asthma, and urticaria, but we have not yet investigated whether LAT mice have allergic diseases. Furthermore, in humans, IgG4-RD involves HLA-DRB1 and FCGR2B at the general level (Reference 15), but whether it is related to LAT has not been investigated. We believe that this is also something that needs further consideration in the future, and this manuscript did not consider it to that extent. 

2. The authors briefly mention the Th2 nature of their observations and the potential link with allergic disease, but there is no follow-up on this in either the experiments or in the discussion. Are there differences in IgE in this model?

COMMENT: 2

Thank you for your advice. As shown in P5L3, one of the characteristics of IgG4-RD is the presence of allergic disease, and IgE often increases. In reference 11, the median IgE of IgG4-related lung disease in 18 patients was 367 (34-2560) U/mL. In addition, IgE is increased in LAT mice (Reference 18). In this study, we revealed that Th2 cytokines are also predominant in the lungs. We believe that this model, including our previous reports, represents the clinical conditions of IgG4-RD. We did not mention it this time, but we will look into presence of allergic disease in LAT mice in future studies. 

3. Figure 1 needs more detail in the results and figure legend, especially regarding the 20 weeks sections. It seems that the image on the left is H&E staining and the image on the right is not, but this is unclear and the stains for each section needs to be defined.

COMMENT: 3

Thank you for your comment. The image on the right is also H-E staining. The left lung was bleeding, so it looks different from the right lung, but it was similar in terms of bronchodilation etc.

4. The Ashcroft score needs to be defined.

COMMENT: 4

We have added the requested information to the text (P10L6).

5. In general, Figures 1-3 would benefit from greater details in the results section.

COMMENT: 5

We have added more details to the text describing Figure 1-3.

Reviewer #2: Dr. Wasada et al present a well argued manuscript describing the use of a LAT knock-in mouse as a model of IgG4-RD of the lung. This is a well executed follow-up study of their prior work, published in this journal, establishing this same mouse model in other organs commonly affected by human IgG4 disease. While the authors provide a compelling argument, there are a few minor issues that I suspect, if addressed, would improve the author's argument.

1) Figure 1: you present two LAT 20w x20 figures at the bottom with largely different appearances. Does this speak to the heterogeneity of the results at 20w? If so, would it help to show cumulative data, compiling the fibrosis scores. If this is true heterogeneity, it may be beneficial to add actual datapoints to Figure 3B, 4, and 5 instead of just showing the median.

COMMENT: 1)

Thank you for your advice. The image on the left looks different because the lung was bleeding, but it is similar if you disregard the effect of bleeding. Because some mice bleed during fibrosis progression, we showed some lungs that are bleeding and some that are not.

2) Redundancy: the statement, "In addition, in Lat knock in mice, the score significantly increased over time." This is already shown in the sentence above, just with data re-arranged. I would re-word or eliminate this.

COMMENT: 2)

Thank you for your advice. The previous sentence shows that the Ashcroft score of LAT mice is significantly higher than that of WT mice after 6 weeks, and the next sentence, "In addition, in Lat knock in mice, the score significantly increased over time.", is a comparison between LAT mice and states that they are different over time. 

3) I'm assuming you chose to study BALF at 8,9 weeks due to your prior results suggesting peak inflammation occurred between 6-10 weeks. I wonder if you would have found greater cytokine production at an earlier time-point. It would have been interesting to see the composition of the lymphocytes found in the BALF. This could be done fairly easily using flow cytometry and could provide some insight into the unique cytokine pattern you found with bead array.

COMMENT: 3)

Thank you for your fruitful advice. Unfortunately, we have not examined cytokines earlier than 8-9 weeks, but in future experiments, we will use flow cytometry to examine the temporal changes in cytokines.

4) Next steps. Now that you have established this model, how do you intend to study it further?

COMMENT: 4)

Thank you for your comment. IgG4-RD is considered to be an allergic disease. Therefore, we would like to examine whether LAT mice have any allergic diseases and evaluate whether inflammation and fibrosis are improved by using antibody preparations such as anti-IL-4, IL-5, and IL-13 antibodies. I think these will be effective for the treatment of IgG4-RD.

5) Wording. There are some phrases throughout the manuscript that are hard to follow, or do not seem to add anything to your argument. For example, on page two, "Regarding the lungs, in addition to comprehensive diagnostic criteria, diagnostic criteria unique to the lungs were established, and diagnosis of lung lesions is performed based on the clinical characteristics." The clarity of your message could be improved greatly by a grammatical review focused on reducing redundancy.

COMMENT: 5)

Thank you for your comments. Regarding the lung, IgG4-positive plasma cells may infiltrate the lung even in lungs with collagen disease (we added reference 12), and patients with diseases other than IgG4-related lung disease will be included based on comprehensive diagnostic criteria alone. Therefore, we established diagnostic criteria unique to the lungs. We rewrote the text to suggest that lung lesions can be accurately diagnosed based on clinical findings, pathological findings, and imaging findings. We also had the entire sentence proofread by a native English-speaking scientist.

---

## [Decision Letter · Decision Letter 1]

3 Feb 2021

The pronounced lung lesions developing in LATY136F knock-in mice mimic human IgG4-related lung disease

PONE-D-20-29792R1

Dear Dr. Waseda,

We’re pleased to inform you that your manuscript has been judged scientifically suitable for publication and will be formally accepted for publication once it meets all outstanding technical requirements.

Kind regards,

Michal A Olszewski, DVM, PhD

Academic Editor

PLOS ONE

Additional Editor Comments (optional):

Reviewers' comments:

Reviewer's Responses to Questions

**Comments to the Author**

1. If the authors have adequately addressed your comments raised in a previous round of review and you feel that this manuscript is now acceptable for publication, you may indicate that here to bypass the “Comments to the Author” section, enter your conflict of interest statement in the “Confidential to Editor” section, and submit your "Accept" recommendation.

Reviewer #2: All comments have been addressed

2. Is the manuscript technically sound, and do the data support the conclusions?

Reviewer #2: Yes

3. Has the statistical analysis been performed appropriately and rigorously? 

Reviewer #2: Yes

4. Have the authors made all data underlying the findings in their manuscript fully available?

Reviewer #2: Yes

5. Is the manuscript presented in an intelligible fashion and written in standard English?

Reviewer #2: Yes

6. Review Comments to the Author

Reviewer #2: The authors were able to partly answer my comments, but I believe sufficiently to make the manuscript acceptable for publication. It is a bit worrisome when individual datapoints are not shown in figures, especially when analyzing mouse data. This was one of my comments that was not addressed and this remains my only reservation.

7. PLOS authors have the option to publish the peer review history of their article (what does this mean?). If published, this will include your full peer review and any attached files.

Reviewer #2: No

---

## [Editor Report · Acceptance letter]

11 Feb 2021

PONE-D-20-29792R1 

The pronounced lung lesions developing in *LAT^Y136F^*knock-in mice mimic human IgG4-related lung disease 

Dear Dr. Waseda:

I'm pleased to inform you that your manuscript has been deemed suitable for publication in PLOS ONE. Congratulations! Your manuscript is now with our production department. 

Kind regards, 

on behalf of

Dr. Michal A Olszewski 

Academic Editor

PLOS ONE